# PNAT: Non-autoregressive Transformer by Position Learning

## Abstract

Non-autoregressive models are promising on various text generation tasks. Previous work hardly considers to explicitly model the positions of generated words. However, the position modeling is an essential problem in non-autoregressive text generation. In this study, we propose PNAT, which incorporates positions as a latent variable into the text generative process. Experimental results show that PNAT achieves top results on machine translation and paraphrase generation tasks, outperforming several strong baselines.

## 1 Introduction

Transformer (Vaswani et al., 2017) has been widely used in many text generation tasks, which is first proposed in neural machine translation, achieving great success for its promising performance. Nevertheless, the auto-regressive property of Transformer has been a bottleneck. Specifically, the decoder of Transformer generates words sequentially, and the latter words are conditioned on previous ones in a sentence. Such bottleneck prevents the decoder from higher efficiency in parallel computation, and imposes strong constrains in text generation, with which the generation order has to be left to right (or right to left) (Shaw et al., 2018; Vaswani et al., 2017).

Recently, many researches (Gu et al., 2018; Lee et al., 2018; Wang et al., 2019; Wei et al., 2019) are devoted to break the auto-regressive bottleneck by introducing non-autoregressive Transformer (NAT) for neural machine translation, where the decoder generates all words simultaneously instead of sequentially. Intuitively, NAT abandons feeding previous predicted words into decoder state at the next time step, but directly copy encoded representation at source side to the decoder inputs (Gu et al., 2018). However, without the auto-regressive constrain, the search space of the output sentence becomes larger (Wei et al., 2019), which brings the performance gap (Lee et al., 2018) between NAT and auto-regressive Transformer (AT). Related works propose to include some inductive priors or learning techniques to boost the performance of NAT. But most of previous work hardly consider explicitly modeling the position of output words during text generation.

We argue that position prediction is an essential problem of NAT. Current NAT approaches do not explicitly model the position of output words, and may ignore the reordering issue in generating output sentences. Compared to machine translation, the reorder problem is much more severe in tasks such as table-to-text (Liu et al., 2018) and dialog generations (Shen et al., 2017). Additionally, it is straightforward to explicitly model word positions in output sentences, as position embeddings are used in Transformer, which is natively non-autoregressive, to include the order information. Intuitively, if output positions are explicitly modeled, the predicted position combined with Transformer to realize non-autoregressive generation would become more natural.

In this paper, we propose non-autoregressive transformer by position learning (PNAT). PNAT is simple yet effective, which explicitly models positions of output words as latent variables in the text generation. Specifically, we introduce a heuristic search process to guide the position learning, and max sampling is adopted to inference the latent model. The proposed PNAT is motivated by learning syntax position (also called syntax distance). Shen et al. (2018) show that syntax position of words in a sentence could be predicted by neural networks in a non-autoregressive fashion, which even obtains top parsing accuracy among strong parser baselines. Given the observations above, we try to directly predict the positions of output words to build a NAT model for text generation.

Our proposed PNAT takes following advantages:

- We propose PNAT, which first includes positions of output words as latent variables for text generation. Experiments show that PNAT achieves very top results in non-autoregressive NMT, outperforming many strong baselines. PNAT also obtains better results than AT in paraphrase generation task.
- Further analysis shows that PNAT has great potentials. With the increase of position prediction accuracy, performances of PNAT could increase significantly. The observations may shed light on the future direction of NAT.
- Thanks to the explicitly modeling of position, we could control the generation by facilitating the position latent variable, which may enable interesting applications such as controlling one special word left to another one. We leave this as future work.

## 2 BACKGROUND

### 2.1 AUTOREGRESSIVE DECODING

A target sequence $Y=y_{1:M}$ is decomposed into a series of conditional probabilities autoregressively, each of which is parameterized using neural networks. This approach has become a de facto standard in language modeling(Sundermeyer et al., 2012), and has been also applied to conditional sequence modeling $p(Y|X)$ by introducing an additional conditional variable $X=x_{1:N}$:

$$p(Y|X) = \prod_{t=1}^{M} p(y_t|y_{<t}, X; \theta) \quad (1)$$

With different choices of neural network architectures such as recurrent neural networks (RNNs) (Bahdanau et al., 2014; Cho et al., 2014), convolutional neural networks (CNNs) (Krizhevsky et al., 2012; Gehring et al., 2017), as well as self-attention based transformer (Vaswani et al., 2017), the autoregressive decoding has achieved great success in tasks such as machine translation (Bahdanau et al., 2014), paraphrase generation (Gupta et al., 2018), speech recognition (Graves et al., 2013), etc.

### 2.2 NON-AUTOREGRESSIVE DECODING

Autoregressive model suffers from the issue of slow decoding in inference, because tokens are generated sequentially and each of them depends on previous ones. As a solution to this issue, Gu et al. (2018) proposed Non-Autoregressive Transformer (denoted as NAT) for machine translation, breaking the dependency among the target tokens through time by decoding all the tokens simultaneously. Put simply, NAT (Gu et al., 2018) factorizes the conditional distribution over a target sequence into a series of conditionally independent distributions with respect to time:

$$p(Y|X) = p_L(M|X : \theta) \cdot \prod_{t=1}^{M} p(y_t|X) \quad (2)$$

which allows trivially finding the most likely target sequence by $\arg\max_Y \; p(Y|X)$ for each timestep $t$, effectively bypassing computational overhead and sub-optimality in decoding from an autoregressive model.

Although non-autoregressive models achieves $15\times$ speedup in machine translation compared with autoregressive models, it comes at the expense of potential performance degradation (Gu et al., 2018). The degradation results from the removal of conditional dependencies within the decoding sentence($y_t$ depend on $y_{<t}$). Without such dependencies, the decoder is hard to leverage the inherent sentence structure in prediction.

### 2.3 LATENT VARIABLES FOR NON-AUTOREGRESSIVE DECODING

A non-autoregressive model could be incorporated with conditional dependency as latent variable to alleviate the degradation resulted from the absence of dependency:

$$P(Y|X) = \int_{\boldsymbol{z}} P(\boldsymbol{z}|X) \prod_{t=1}^{M} P(y_t|\boldsymbol{z}, X) dz \quad (3)$$

For example, NAT-FT (Gu et al., 2018) models the inherent sentence structure with a latent fertility variable, which represents how many target tokens that a source token would translate to. Lee et al. (2018) introduces $L$ intermediate predictions $Y^{1:L}$ as random variables , and to refine the predictions from $Y^1$ to $Y^L$ in a iterative manner.

# 3 PNAT: POSITION-BASED NON-AUTOREGRESSIVE TRANSFORMER

We propose position-based non-autoregressive transformer (*PNAT*), an extension to transformer incorporated with non auto-regressive decoding and position learning.

## 3.1 MODELING POSITION WITH LATENT VARIABLES

Languages are usually inconsistent with each other in word order. Thus reordering is usually required when translating a sentence from a language to another. In NAT family, words representations or encoder states at source side are copied to the target side to feed into decoder as its input. Previously, Gu et al. (2018) utilizes positional attention which incorporates positional encoding into decoder attention to perform local reordering. But such implicitly reordering mechanism by position attention may cause a *repeated generation* problem, because position learning module is not optimized directly, and is likely to be misguided by target supervision.

To tackle with this problem, we propose to explicitly model the position as a latent variable. We rewrite the target sequence $Y$ with its corresponding position latent variable $\boldsymbol{z} = z_{1:M}$ as a set $Y_{\boldsymbol{z}} = y_{z_1:z_M}$. The conditional probability $P(Y|X)$ is factorized with respect to the position latent variable:

$$P(Y|X) = \sum_{\boldsymbol{z} \in \pi(M)} P(\boldsymbol{z}|X) \cdot P(Y|\boldsymbol{z}, X) \tag{4}$$

where $\pi(M)$ is a set consisting of permutations with $M$ elements. At decoding time, the factorization allows us to decode sentences in parallel by pre-predicting the corresponding position variables $z$.

## 3.2 MODEL ARCHITECTURE

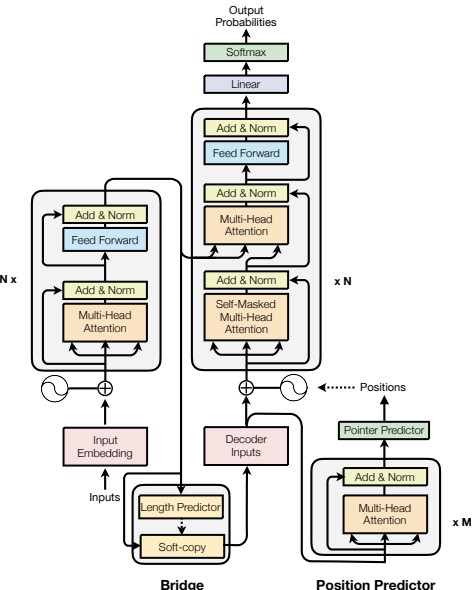

Figure 1: Illustration of the proposed model, where the black solid arrows represent differentiable connections and the dashed arrows are non-differentiable operations.

As shown in Figure 1, PNAT is composed of four modules: an encoder stack, a bridge block, a position predictor as well as a decoder stack. Before detailing each component of PNAT model, we overview the architecture for a brief understanding.

Like most sequence-to-sequence models, PNAT first encodes a source sequence $X=x_{1:N}$ into its contextual word representations $E=e_{1:N}$ with the encoder stack. With generated contextual word representation $E$ at source side, the bridge block is leveraged to computed the target length $M$ as well as the corresponding features $D=d_{1:M}$, which is fed into the decoder as its input. It is worth noting that the decoder inputs $D$ is computed without reordering. Thus the position predictor is introduced to deal with this issue by predicting a permutation $z=z_{1:M}$ over $D$. Finally, PNAT generates the target sequence from the decoder input $D$ and its permutation $z$.

**Encoder and Decoder** Given a source sentence $X$ with length $N$, PNAT encoder produces its contextual word representations $E$. The contextual word representations $E$ are further used in computing target length $M$ and decoder initial states $D$, and are also used as memory of attention at decoder side.

Generally, PNAT decoder can be considered as a transformer with a broader vision, because it leverages future word information that is blind to the autoregressive transformer. Intuitively, we use relative position encoding in self-attention(Shaw et al., 2018), rather than absolute one that is more likely to cause position errors. Following Shaw et al. (2018) with a clipping distance $d$ (usually $d \geq 2$) set for relative positions, we preserve $d = 4$ relations.

**Bridge** The bridge module predicts the target length $M$, and initializes the decoder inputs $D$ from the source representations $E$. The target length $M$ could be estimated from the source encoder representation:

$$M = N + \arg\max \phi(E) \tag{5}$$

where $\phi(\cdot)$ produces a categorical distribution ranged in $[-B, B]$ ($B = 20$). It is notable that we use the predicted length at inference stage, although during training, we simply use the length of each reference target sequence. Then, we adopt the method proposed by Li et al. (2019) to compute $D$. Given the source representation $E$ and the estimated target length $M$, we linearly combine the embeddings of the neighboring source tokens to generate $D$ as follows:

$$d_j = \sum_i w_{ji} \cdot e_i \tag{6}$$

$$w_{ji} = \text{softmax}(-|j - i|/\tau) \tag{7}$$

where $w_{ji}$ is a normalized weight that reflects the contribution of $e_i$ to $d_j$, and $\tau$ is a hyperparameter indicating the sharpness of the weight distribution.

**Position Predictor** For the proposed PNAT, we model position permutations with a position predictor. As shown in Figure 1, the position predictor takes the decoder inputs $D$ and the source representation $E$ to predict a permutation $z$. The position predictor has a sub-encoder which stacks multiple layers of encoder units to predict its predicted input $R=r_{1:M}$.

With the predicted inputs $R$, we conduct an autoregressive position predictor, denoted as *AR-Predictor*. The AR-Predictor searches a permutation $z$ with:

$$P(z|D, E) = \prod_{t=1}^{M} p_(z_t|z_{<t}, D, E; \theta) \tag{8}$$

where $\theta$ is the parameter of AR-Predictor, which includes a RNN-based model incorporated with a pointer network (Vinyals et al., 2015).

To purse the efficiency of decoding, we also explore a non-autoregressive version for the position predictor, denoted as NAR-Predictor, to model the position permutation probabilities with:

$$P(z|D, E) = \prod_{t=1}^{M} p(z_t|D, E; \theta) \tag{9}$$

To obtain the permutation $z$, AR-Predictor performs greedy search whereas NAR-Predictor performs direct $\arg\max$. We chose the AR-Predictor as our mainly position module in PNAT, and we also analyze the effectiveness of position modeling in Sec. 4.4.

### 3.3 TRAINING

Training requires maximizing the marginalized likelihood in Eqn. 4. However, this is intractable since we need to enumerate all the $M!$ permutations of tokens. We therefore optimize this objective by Monte Carlo sampling method with a heuristic search algorithm.

**Heuristic Search for Positions**    Intuitively, each target token should have a corresponding decoder input, and meanwhile each decoder input should be assigned to a target token. Based on this idea, we design a heuristic search algorithm to allocate positions. Given the decoder inputs and its target tokens, we first estimate the similarity between each pair of the decoder input $d_i$ and the target token embedding $y_j$, which is also the weights of the target word classifier:

$$\text{sim}_{i,j} = \text{cosine}\,(d_i, y_j) \tag{10}$$

Based on the cosine similarity matrix, HSP is designed to find a perfect matching between decoder inputs and target tokens:

$$\text{HSP}(\boldsymbol{z}) = \arg\max_{\boldsymbol{z}} \sum_{i=0}^{M} (\text{sim}_{i,\boldsymbol{z}_i}) \tag{11}$$

Here we apply a greedy algorithm to select the pair with the highest similarity score iteratively until a permutation $\boldsymbol{z}_{\text{ref}}$ is generated. More details are provided in Appendix A.

The intuition behind is that, if the decoder input $d_i$ is already the most similar one to a target word, it would be easier to keep and even reinforce this association in learning the model. We also analyze the effectiveness of the HSP in the Sec. 4.4.

**Objective Function**    With the heuristically discovered positions as reference positions $\boldsymbol{z}_{\text{ref}}$, the position predictor could be trained with a position loss:

$$\mathcal{L}_{\text{p}} = -\log P(\boldsymbol{z}_{\text{ref}}|D, E) \tag{12}$$

Grounding on the referenced positions, the generative process of target sequences is optimized by:

$$\mathcal{L}_{\text{g}} = -\sum_{t=1}^{M} \log P(Y|\boldsymbol{z}_{\text{ref}}; X) \tag{13}$$

Finally, combining two loss functions mentioned above, a full-fledged loss is derived as

$$\mathcal{L} = \mathcal{L}_{\text{g}} + \alpha \mathcal{L}_{\text{p}} \tag{14}$$

The length predictor is a classifier that follows the previous settings. We also follow the previous practice (Gu et al., 2018; Wei et al., 2019) and perform an extra training process for the length predictor after the model trained and do not tune the parameter of the encoder.

### 3.4 INFERENCE

We follow the common choice of approximating decoding algorithms (Gu et al., 2018; Lee et al., 2018) to reduce the search space of latent variable model.

**Argmax Decoding**    Following Gu et al. (2018), one simple and effective method is to select the best sequence by choosing the highest-probability latent sequence $z$:

$$\boldsymbol{z}^* = \arg\max_{\boldsymbol{z}} P(\boldsymbol{z}|D, E)$$
$$Y^* = \arg\max_{y} P(Y|\boldsymbol{z}^*, X)$$

where identifying $Y^*$ only requires independently maximizing the local probability for each output position.

**Length Parallel Decoding**    We also consider the common practice of noisy parallel decoding (Gu et al., 2018), which generates a number of decoding candidates in parallel and selects the best via re-scoring using a pre-trained autoregressive model. For PNAT, we first predict the target length as $\hat{M}$, then generate output sequence with argmax decoding for each target length candidate $M \in [\hat{M} - \Delta M, \hat{M} + \Delta M]$ ($M = 4$ in our experiments), which was called length parallel decoding (LPD). Then we use the pre-trained autoregressive model to rank these sequences and identify the best overall output as the final output.

## 4    EXPERIMENTS

We test PNAT on several benchmark sequence generation tasks. We first describe the experimental setting and implementation details and then present the main results, followed by some deep studies.

### 4.1    EXPERIMENTAL SETTING

To show the generation ability of PNAT, we conduct experiments on the popular machine translation and paraphrase generation tasks. These sequence generation task evaluation models from different perspectives. Translation tasks test the ability of semantic transforming across bilingual corpus. While paraphrase task focuses on substitution between the same languages while keeping the semantics.

**Machine Translation**    We valid the effectiveness of PNAT on the most widely used benchmarks for machine translation — WMT14 EN-DE(4.5M pairs) and IWSLT16 DE-EN(196K pairs). The dataset is processed with Moses script (Koehn et al., 2007), and the words are segmented into subword units using byte-pair encoding (Sennrich et al., 2016, BPE). For both WMT datasets, the source and target languages share the same set of subword embeddings while for IWSLT we use separate embeddings.

**Paraphrase Generation**    We conduct experiments following previous work (Miao et al., 2019) for paraphrase generation. We make use of the established Quora dataset [1] to evaluate on the paraphrase generation task. We consider the supervised paraphrase generation and split the Quora dataset in the standard setting. We sample 100k pairs sentence as training data, and holds out 3k, 30k for validation and testing, respectively.

### 4.2    IMPLEMENTATION DETAILS

**Module Setting**    For machine translation, we follow the settings from Gu et al. (2018). In the case of IWSLT task, we use a small setting ($d_{\text{model}} = 278$, $d_{\text{hidden}} = 507$, $p_{\text{dropout}} = 0.1$, $n_{\text{layer}} = 5$ and $n_{\text{head}} = 2$) suggested by Gu et al. (2018) for Transformer and NAT models. For WMT task, we use the base setting of the Vaswani et al. (2017) ($d_{\text{model}} = 512$, $d_{\text{hidden}} = 512$, $p_{\text{dropout}} = 0.1$, $n_{\text{layer}} = 6$).

For paraphrase generation, we follow the settings from Miao et al. (2019), and set the 300-dimensional GRU with 2 layer for Seq-to-Seq (GRU). We empirically select a Transformer and NAT models with hyperparameters ($d_{\text{model}} = 400$, $d_{\text{hidden}} = 800$, $p_{\text{dropout}} = 0.1$, $n_{\text{layer}} = 3$ and $n_{\text{head}} = 4$).

**Optimization**    We optimize the parameter with the Adam optimizer (Kingma & Ba, 2014). The hyperparameter $\alpha$ used in Eqn. 14 was be set to 1.0 for WMT, 0.3 for IWSLT and Quora. We also use inverse square root learning rate scheduling (Vaswani et al., 2017) for the WMT, and using linear annealing (from $3e - 4$ to $1e - 5$, suggested by Lee et al. (2018)) for the IWSLT and Quora. Each mini-batch consists of approximately 2K tokens for IWSLT and Quora, 32K tokens for WMT.

**Knowledge Distillation**    Sequence-level knowledge distillation is applied to alleviate multi-modality problem while training, using Transformer as a teacher (Hinton et al., 2015). Previous studies on non-autoregressive generation (Gu et al., 2018; Lee et al., 2018; Wei et al., 2019) have used translations produced by a pre-trained Transformer model as the training data, which significantly improves the performance. We follow this setting in translation tasks.

---

[1]https://www.kaggle.com/c/quora-question-pairs/data

| Model | WMT 14 | | IWSLT16 | Speedup |
| | EN-DE | DE-EN | DE-EN | |
|---|---|---|---|---|
| Autoregressive Methods | | | | |
| Transformer-base (Vaswani et al., 2017) | 27.30 | / | / | / |
| *Transformer(Beam=4) | 27.40 | 31.33 | 34.81 | 1.0× |
| Non-Autoregressive Methods | | | | |
| Flowseq (Ma et al., 2019) | 18.55 | 23.36 | / | / |
| *NAT-base | / | 11.02 | / | / |
| *PNAT | **19.73** | **24.04** | / | / |
| NAT w/ Knowledge Distillation | | | | |
| NAT-FT (Gu et al., 2018) | 17.69 | 21.47 | / | 15.6× |
| LT (Kaiser et al., 2018) | 19.80 | / | / | 5.8× |
| IR-NAT (Lee et al., 2018) | 13.91 | 16.77 | 27.68 | 9.0× |
| ENAT (Guo et al., 2019) | 20.65 | 23.02 | / | 24.3× |
| NAT-REG (Wang et al., 2019) | 20.65 | 24.77 | / | - |
| imitate-NAT (Wei et al., 2019) | 22.44 | 25.67 | / | 18.6× |
| Flowseq (Ma et al., 2019) | 21.45 | 26.16 | / | 1.1× |
| *NAT-base | / | 16.69 | / | 13.5× |
| *PNAT | **23.05** | **27.18** | **31.23** | 7.3 × |
| NAT w/ Reranking or Iterative Refinments | | | | |
| NAT-FT (rescoring 10 candidates) | 18.66 | 22.42 | / | 7.7× |
| LT (rescoring 10 candidates) | 22.50 | / | / | / |
| IR-NAT (refinement 10) | 21.61 | 25.48 | 32.31 | 1.3× |
| ENAT (rescoring 9 candidates) | 24.28 | 26.10 | / | 12.4× |
| NAT-REG (rescoring 9 candidates) | **24.61** | 28.90 | / | - |
| imitate-NAT (rescoring 9 candidates) | 24.15 | 27.28 | / | 9.7× |
| Flowseq (rescoring 30 candidates) | 23.48 | 28.40 | / | / |
| *PNAT (LPD $n$=9,$\Delta M$=4) | 24.48 | **29.16** | **32.60** | 3.7× |

Table 1: Performance on the newstest-2014 for WMT14 EN-DE and test2013 for IWSLT EN-DE. '-' denotes same numbers as above. '*' indicates our implementation. The decoding speed is measured sentence-by-sentence and the speedup is computed by comparing with Transformer.

## 4.3 MAIN RESULTS

**Machine Translation**  We compare the PNAT with strong NAT baselines, including the NAT with fertility (Gu et al., 2018, NAT-FT), the NAT with iterative refinement (Lee et al., 2018, IR-NAT), the NAT with regularization (Wang et al., 2019, NAT-REG), the NAT with enhanced decoder input (Guo et al., 2019, ENAT), the NAT with learning from auto-regressive model (Wei et al., 2019, imitate-NAT), the NAT build on latent variables (Kaiser et al., 2018, LT), and the flow-based NAT model (Ma et al., 2019, Flowseq).

The results are shown in Table 1. We basically compare the proposed PNAT against the autoregressive counterpart both in terms of generation quality, which is measured with BLEU (Papineni et al., 2002) and inference speedup. For all our tasks, we obtain the performance of competitors by either directly using the performance figures reported in the previous works if they are available or producing them by using the open source implementation of baseline algorithms on our datasets.[2] Clearly, PNAT achieves a comparable or better result to previous NAT models on both WMT and IWSLT tasks.

We list the result of the NAT models trained without using knowledge distillation in the second block of the Table 1. The PNAT achieves significant improvements (more than 13.0 BLEU points) over the naive baselines, which indicate that position learning greatly contributes to improve the model capability of NAT model. The PNAT also achieves a better result than the Flowseq around 1.0 BLEU, which demonstrates the effectiveness of PNAT in modeling dependencies between the target outputs.

As shown in the third block of the Table 1, without using reranking techniques, the PNAT outperforms all the competitors with a large margin, achieves a balance between performance and efficiency. In

---

[2]For the sake of fairness, we have chosen the base setting for all competitors.

particular, the previous state-of-the-art(WMT14 DE-EN) Flowseq achieves good performance with the slow speed($1.1\times$), while PNAT goes beyond Flowseq in both respects.

Our best results are obtained with length parallel decoding which employ autoregressive model to rerank the multiple generation candidates of different target length. Specifically, on the large scale WMT14 DE-EN task, PNAT (+LPD) surpass the NAT-REG by 0.76 BLEU score. Without reranking, the gap has increased to 2.4 BLEU score (27.18 v.s. 24.77). The experiments shows the power of explicitly position modeling which reduces the gap between non-autoregressive and the autoregressive models.

**Paraphrase Generation**  Given a sentence, paraphrase generation aims to synthesize another sentence that is different from the given one, but conveys the same meaning. Comparing with translation task, paraphrase generation prefers a more similar order between source and target sentence, which possibly learn a trivial position model. PNAT can potentially yield better results with the position model to infer the relatively ordered alignment relationship.

| Model | Paraphrase(BLEU) | |
|---|---|---|
| | Valid | Test |
| Seq-to-seq(GRU) | 24.68 | 24.75 |
| Transformer | 25.88 | 25.46 |
| NAT-base | 19.80 | 20.34 |
| PNAT | **29.30** | **29.00** |

Table 2: Results on validation set and test set of Quora.

The results of the paraphrase generation are shown in Table 2. In consist with our intuition, PNAT achieves the best result on this task and even surpass Transformer around 3.5 BLEU. The NAT model is not powerful enough to capture the latent position relationship. The comparison between NAT-base and PNAT shows that explicit position modeling in PNAT plays a crucial role in generating sentences.

## 4.4 ANALYSIS

**Effectiveness of Heuristic Searched Position**  First, we analyze whether the position derived from the heuristic search is suitable for use as supervision to the position predictor. We evaluate the effectiveness of the searched position by training a PNAT as before and testing with the heuristic searched position instead of the predicted position. As shown in the second block of the Table 3, it is easier noticed that as PNAT w/ HSP achieves a significant improvement over the NAT-base and the Transformer, which demonstrates that the heuristic search for the position is effective.

| Model | Position Accuracy(%) | | WMT14 DE-EN BLEU | Speed Up |
|---|---|---|---|---|
| | permutation-acc | relative-acc(r=4) | | |
| Transformer(beam=4) | / | / | 30.68 | 1.0$\times$ |
| NAT-base | / | / | 16.71 | 13.5$\times$ |
| PNAT w/ HSP | 100.00 | 100.00 | 46.03 | 12.5$\times$ |
| PNAT w/ AR-Predictor | 25.30 | 59.27 | 27.11 | 7.3$\times$ |
| PNAT w/ NAR-Predictor | 23.11 | 55.57 | 20.81 | 11.7$\times$ |

Table 3: Results on validation set of WMT14 DE-EN with different position strategy. "HSP" means the reference position sequence derived from the heuristic position searching.

**Effectiveness and Efficiency of Position Modeling**  We are also analysis the accuracy of our position modeling and its influence on the quality of generation on the WMT14 DE-EN task. For evaluating the position accuracy, we adopt the heuristic searched position as the position reference (denoted as "HSP"), which is the training target of the position predictor. PNAT requires the position information at two places. The first is the mutual relative relationship between the states that will be used during decoding. And the second is to reorder the decoded output after decoding. We

then propose the corresponding metrics for evaluation, which is the relative position accuracy (with relation threshold $r = 4$) and the permutation accuracy.

As shown in Table 3, better position accuracy always yields better generation performance. The non-autoregressive position model is less effective than the current autoregressive position model, both in the accuracy of the permutation and the relative position. Even though the current PNAT with a simple AR-Predictor has surpassed the previous NAT model, the position accuracy is still less desirable (say, less than 30%) and has a great exploration space. We provide a few examples in Appendix B. There is also a trade-off between the effectiveness and efficiency, the choice of the non-autoregressive means the efficiency and the choice of autoregressive means the effectiveness.

**Repeated Generation Analysis** Previous NAT often suffers from the repeated generation problem due to the lack of sequential position information. NAT is less effective to distinguish adjacent decoder hidden states, which is copied from the adjacent source representation. To further study this problem, we proposed to evaluate the gains of simply remove the repeated tokens. As shown in Table 4, we perform the repeated generation analysis on the paraphrase generation tasks. Removing repeated tokens has little impact for PNAT model, with only 0.05 BLEU differences. However for the NAT-base model, the gap comes with almost 1 BLEU (0.89). The results clearly demonstrate that the explicitly position model essentially learns the sequential information for sequence generation.

| Model | Paraphrase(Test-BLEU) | | |
| --- | --- | --- | --- |
| | w/ remove repeats | w/o remove repeats | $\Delta_{\text{BLEU}}$ |
| NAT-base | 20.34 | 19.45 | 0.89 |
| PNAT | 29.00 | 28.95 | 0.05 |

Table 4: Results on test set of Quora.

**Convergence Efficiency** We also perform the training efficiency analysis in IWSLT16 DE-EN Translation task. The learning curves are shown in 2. The curve of the PNAT is on the top-left corner. Remarkably, PNAT has the best convergence speed compared with the NAT competitors and even a strong autoregressive model. The results are in line with our intuition, that the position learning brings meaningful information of position relationship and benefits the generation of the target sentence.

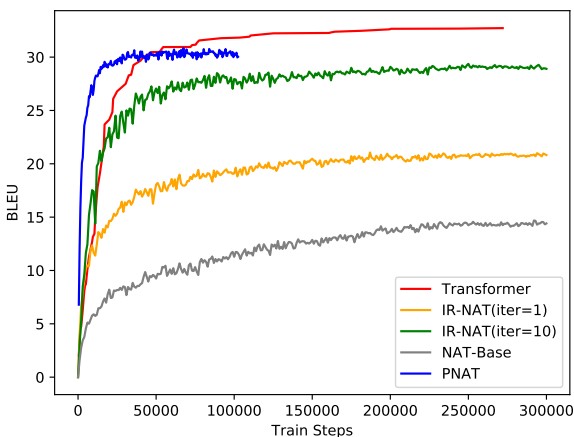

Figure 2: The learning curves from training of models on evaluation set of IWSLT-16 DE-EN. Mini-batch size is 2048 tokens.

## 5 RELATED WORK

Gu et al. (2018) first develops a non-autoregressive Transformer for neural machine translation (NMT) tasks, which produces the outputs in parallel and the inference speed is thus significantly boosted.

Due to the removal of the dependencies between the target outputs, it comes at the cost that the translation quality is largely sacrificed. A line of work has been proposed to mitigate such performance degradation. Some previous work is focused on enhancing the decoder inputs by replacing the target words as inputs, such as Guo et al. (2019) and Lee et al. (2018). Lee et al. (2018) proposed a method of iterative refinement based on the latent variable model and denoising autoencoder. Guo et al. (2019) enhances decoder input by introducing the phrase table in statistical machine translation and embedding transformation. Another part of previous work focuses on improving the supervision of NAT's decoder states, including imitation learning from autoregressive models (Wei et al., 2019) or regularizing the decoder state with backward reconstruction error (Wang et al., 2019). There is also a line studies build upon latent variables, such as Kaiser et al. (2018) and Roy et al. (2018) utilize discrete latent variables for making decoding more parallelizable. Moreover, Shao et al. (2019) also proposed a method to retrieve the target sequential information for NAT models. Unlike previous work, we explicitly model the position, which has shown its importance to the autoregressive model and can well model the dependence between states. To the best of our knowledge, PNAT is the first work to explicitly model position information for non-autoregressive text generation.

## 6 CONCLUSION

We proposed PNAT, a non-autoregressive transformer by explicitly modeled positions, which bridge the performance gap between the non-autoregressive decoding and autoregressive decoding. Specifically, we model the position as latent variables, and training with heuristic searched positions with MC algorithms. As a result, PNAT leads to significant improvement and move more close to the performance gap between the NAT and AT on machine translation tasks. Besides, the experimental results of the paraphrase generation task show that the performance of the PNAT can exceed that of the autoregressive model, and at the same time, it also has a large improvement space. According to our further analysis on effectiveness of position modeling, in future work, we can still enhance the performance of the NAT model by strengthening position learning.

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

# A HEURISTIC SEARCH FOR POSITIONS

---

**Algorithm 1** Heuristic Search for Positions

---

**Input:** The candidates set of decoder inputs: $D = \{d_1, \cdots, d_M\}$ and target embeddings: $Y = \{y_1, \cdots, y_M\}$;

**Output:** The position of the decoder inputs $\hat{z}$.

1: initial $A = \{\}$, $\hat{D} = D$, $\hat{Y} = Y$;
2: compute the similarity matrix $\mathrm{Sim}_{D,Y}$: the $\mathrm{Sim}[i,j]$ in the matrix is the the similarity between the $d_i$ and $y_j$ computing with $\mathrm{sim}_{i,j} = \mathrm{cosine}\,(d_i, y_j)$;
3: **repeat**
4:      extract the similarity matrix $\mathrm{Sim}_{\hat{D},\hat{Y}}$ from the $\mathrm{Sim}_{D,Y}$;
5:      select: $(i,j) = \arg\max_{(i,j)} \mathrm{Sim}_{\hat{D},\hat{Y}}$
6:      update: $A \leftarrow A \cup \{(i,j)\}$, $\hat{D} \leftarrow \hat{D} \setminus \{d_i\}$, $\hat{Y} \leftarrow \hat{Y} \setminus \{y_j\}$;
7: **until** $\hat{D} = \{\}$ and $\hat{Y} = \{\}$
8: **for** each pair $(i,j)$ in $A$ **do**
     set $\hat{z}_i = j$
9: **end for**;
10: return $\hat{z}$

---

As shown in Algorithm 1, we perform a greedy algorithm to select the pair with the highest similarity score iteratively until the permutation $\hat{z}$ is generated.

The complexity of this algorithm is $o(M^3)$ ($M$ is the length of output sentence). Specifically, the complexity to select the maximum from the similarity matrix is $o(M^2)$ for each loop. We need $M$ loops of greedy search to allocate positions for all decoder inputs.

# B CASE STUDY OF PREDICTED POSITIONS

We also provide a few examples in Table 5. For each source sentence, we first analyze the generation quality of the PNAT with a heuristic searched position. Besides, we also show the translation with the predicted position. We have the following observations: First, the output generated by the PNAT using the heuristic searched position always keeps the high consistency with the reference, shows the effectiveness of the heuristic searched position. Second, better position accuracy always yields better generation performance (Case 1,2 against Case 3). Third, as we can see in case 4, though the permutation accuracy is lower, it still generates a good result, the reason why we chose to use the relative self-attention instead of absolute self-attention.

| Source | bei dem deutschen Gesetz geht es um die Zuweisung bei der Geburt . |
|---|---|
| Reference | German law is about assigning it at birth . |
| Heuristic Searched Position(HSP) | 3, 6, 1, 2, 10, 0, 5, 4, 7, 8, 9 |
| PNAT w/ HSP | German law is about assigning them at birth . |
| Predicted Position | 3, 6, 1, 2, 10, 0, 5, 4, 7, 8, 9 |
| PNAT w/ Predicted Postion | German law is about assigning them at birth . |
| Source | weiß er über das Telefon @-@ Hacking Bescheid ? |
| Reference | does he know about phone hacking ? |
| Heuristic Searched Position(HSP) | 2, 1, 3, 4, 8, 5, 6, 0, 7, 9 |
| PNAT w/ HSP | does he know the telephone hacking ? |
| Predicted Position | 1, 0, 3, 4, 8, 5, 6, 2, 7, 9 |
| PNAT w/ Predicted Postion | he know about the telephone hacking ? |
| Source | was CCAA bedeutet , möchte eine Besucherin wissen . |
| Reference | one visitor wants to know what CCAA means . |
| Heuristic Searched Position(HSP) | 5, 6, 7, 8, 9, 3, 2, 0, 1, 11, 4, 10 |
| PNAT w/ HSP | a visitor wants to know what CCAA means . |
| Predicted Position | 5, 0, 1, 2, 3, 7, 4, 8, 9, 11, 6, 10 |
| PNAT w/ Predicted Postion | CCAA means wants to know to a visitor . |
| Source | eines von 20 Kindern in den Vereinigten Staaten hat inzwischen eine Lebensmittelal-lergie . |
| Reference | one in 20 children in the United States now have food allergies . |
| Heuristic Searched Position(HSP) | 14, 1, 2, 3, 4, 5, 6, 7, 9, 8, 0, 10, 11, 12, 13 |
| PNAT w/ HSP | one of 20 children in the United States now has food allergy . |
| Predicted Position | 14, 0, 1, 2, 3, 4, 5, 6, 8, 7, 9, 10, 11, 12, 13 |
| PNAT w/ Predicted Postion | of 20 children in the United States now has a food allergy . |

Table 5: Examples of translation outputs from PNAT with different setting on WMT14 DE-EN. It is should be noted that the length is different between the position sequence and the output sequence because we keep the origin position output and combine the BPE sequence to word sequence.

