# OpenReview forum: "PNAT: Non-autoregressive Transformer by Position Learning"
_ICLR.cc/2020/Conference — Reject_

### Official Review · AnonReviewer1 · 2019-10-22
**Official Blind Review #1**

**Rating:** 6

**Review:**

This work proposes a non-autoregressive model for conditioned text generation. The non-autoregressive decoder conditions on a sequence of discrete latent variables, which represent the generation order and can be autoregressively calculated. Instead of doing marginal inference, the paper takes the top 1 generation order that best match inputs. Experiments on machine translation and paraphrase generation show strong result in comparison to other non-autoregressive models.

The idea of delegating generation order to the latent variables seems interesting, and the paper takes a reasonable approach when fleshing it out. Most of the experiments seem solid to me. The presentation does a good job in giving a very brief description of the model to readers familiar with non-autoregressive generation; but for those who are not (like myself), much content needs to be clarified.

Details:

- How is the model trained? Eq.13--14 present the loss, does it ever include the length predictor? Is the computation graph fully-diiferentiable? If it is, how do the authors backprop through the argmax in the length? If not, is reinforcement learning used?

- Can the authors specify B around Eq.5? Also \Delta M above section 4.3.

- Is the length predictor basically a classifier? Have the authors considered doing regression, which preserves the order relation (e.g., 3 is less than 5)?

- HSP. Can the authors describe the matching algorithm in detail? If it is some well-known algo like the Hungarian algorithm, please specify. If the authors come up with their own algo, please typeset it and analyze the complexity.

- Adding onto the above: is the algorithm exact or approximate? If the later, Eq.11 should not be \argmax.

- Table 1 is really confusing: what do NPD, WT, i_dec stand for?

- 3.2 discusses two position predictors. Which of them is actually used?

- Tabl 2: what does `Searched-Position` stand for? I thought the position search is only used to derived the reference position sequence.

- It is a bit sad that the paper started with marginal inference but ends up taking the top 1. Have the authors considered taking top K of the predicted orders to have a better approximation to the likelihood?


Minor:

- The second paragraph in intro. I'm not sure why non-autoregressive model has a larger search space. Both of them search over \Sigma^\ast, with \Sigma being the vocabulary.

- Typo above section 4.4: ... not powerful enough....

**Experience Assessment:**

I have published one or two papers in this area.

**Review Assessment: Checking Correctness Of Derivations And Theory:**

N/A

**Review Assessment: Checking Correctness Of Experiments:**

I assessed the sensibility of the experiments.

**Review Assessment: Thoroughness In Paper Reading:**

I read the paper at least twice and used my best judgement in assessing the paper.

---

> ### Author Response · Authors · 2019-11-15
> **Response to Reviewer #1**
>
> Thank you for your valuable reviews. We revised accordingly, please check the common response. We have updated our paper and try to address the issues you mentioned in the comment.
>
> Q1: "How is the model trained? Eq.13--14 present the loss, does it ever include the length predictor? Is the computation graph fully-diiferentiable? If it is, how do the authors backprop through the argmax in the length? If not, is reinforcement learning used?"
>
> Response:  The position predictor and word predictor in PNAT are jointly training by optimizing the Eq.14. The length predictor is trained separately as a classifier (Gu et al.[1]). Its predictions are later used in the decoder without updating the classifier. In such a case, BP works well and we do not need to use RL.
>
> Q2: "Can the authors specify B around Eq.5? Also \Delta M above section 4.3."
>
> Response: B=20, \Delta M = 4. We have corrected this in the latest revision.
>
> Q3: "Is the length predictor basically a classifier? Have the authors considered doing regression, which preserves the order relation (e.g., 3 is less than 5)?"
>
> Response:
> - Yes. The length predictor is a classifier that follows the previous settings(Gu et al.[1]).
> - Thanks for your very good suggestion, we will explore this as the future study.
>
> Q4: "HSP. Can the authors describe the matching algorithm in detail? If it is some well-known algo like the Hungarian algorithm, please specify. If the authors come up with their own algo, please typeset it and analyze the complexity."
>
> Response:
> - We do not use the Hungarian algorithm. HSP is specifically designed for position learning in NAT. We have added more details about the algorithm in the appendix. The brief answer is: At each step, the best matching pair of d_i and y_j are selected and removed from the candidates. The algorithm selects the matching pair iteratively until the set of candidates becomes empty.
> - The complexity of the algorithm is o(M^3), where M is the length of the target sentence. We chose this algorithm because it is easier to perform parallel computations in training.
> - We have also conducted the experiments and verified the effectiveness of HSP we used here in Section 4.4. PNAT w/ HSP achieves a significant improvement over the NAT-base and the Transformer, which demonstrates that the heuristic search for the position is very effective.
>
> Q5: "Adding onto the above: is the algorithm exact or approximate? If the later, Eq.11 should not be \argmax."
>
> Response:
> - As described in the R4, it is an approximate algorithm.
> - Argmax means that we perform a greedy search based on the matching of the decoder inputs and the target embedding.
>
> Q6: "Table 1 is really confusing: what do NPD, WT, i_dec stand for?"
>
> Response:
> - Thanks for your reminder and we have corrected this in the latest version.
> - NPD: noisy parallel decoding used for beam searching in previous NATs work, such as Gu et al.[1]
> - WT: weak teacher version of Transformer which has a similar performance to the Transformer used in Gu et al.
> - The i_dec is the number of iterations used for iterative refinements.(Lee et al. [2])
>
> Q7: "3.2 discusses two position predictors. Which of them is actually used?"
>
> Response:
> - We mainly chose the AR-Predictor as the position predictor in our experiments.
> - We also analyze the PNAT using the NAR-Predictor as the position predictor in Section 4.4,   the non-autoregressive position model is less effective than the current autoregressive position model, both in the accuracy of the permutation and the relative position.
>
> Q8: "The second paragraph in intro. I'm not sure why non-autoregressive model has a larger search space. Both of them search over \Sigma^\ast, with \Sigma being the vocabulary."
>
> Response:
> - Larger search space means each decoding state in the NAT models has to decide not only which part of the target sentence it will focus on, but also the correct target word for that part. (Wei et al.[3])
> - The autoregressive models apply sequential decoding to naturally generate successive tokens and do not suffer from the problem above.
> - Therefore, we think that the search space of NAT larger than the AT.
>
> [1] Gu J, Bradbury J, Xiong C, et al. Non-autoregressive neural machine translation. ICLR 2018.
> [2] Lee J, Mansimov E, Cho K. Deterministic non-autoregressive neural sequence modeling by iterative refinement. EMNLP 2018.
> [3] Wei B, Wang M, Zhou H, et al. Imitation Learning for Non-Autoregressive Neural Machine Translation. ACL 2019.

---

### Official Review · AnonReviewer2 · 2019-10-25
**Official Blind Review #2**

**Rating:** 3

**Review:**

This work proposes an alternative approach to non-autoregressive translation (NAT) by predicting positions in addition to the word identities, such that the word order in the final prediction doesn't matter as long as the positions are correct. The length of the translation is predicted similar to Gu et al 2017, as well as smoothly copying the source sequence to decoder input. However, since the positions are unknown, this paper employs a heuristic search method to find the nearest neighbors in the embedding space to obtain position supervision.

Experiments are conducted on a machine translation task and a paraphrase generation task. For WMT14 DE-EN and IWSLT16 DE-EN, this method gets superior performance compared to baselines. For paraphrase generation, this approach with an AR position predictor gets even better performance than a normal AR transformer. Further analysis shows that this approach degrades less without knowledge distillation, or without removing repetitions.

Pros:
1. This approach gets better performance compared to baselines.
2. The idea of modeling positions as a latent variable is interesting and might generalize to other tasks beyond NAT.

Cons:
1. This work should compare to later baselines such as FlowSeq (https://arxiv.org/pdf/1909.02480.pdf) which gets better performance with flow.
2. In table 1, although the proposed approach outperforms imitate-NAT, the speedup is lower, making it hard to judge which is actually better.
3. In table 2, why is AR predictor used? What's the performance of NAR predictor? (in general why consider NAR position predictor at all?)
4. It is not clear why heuristic search would work here. Is any pretraining required? Otherwise, since there's no gradient signal for the positions, I'm not sure how the model figures it out.

Questions:
1. How many samples are used in table 1 LPD? Or is it argmax decoding for each length?
2. Is it possible to include a few examples showing predicted positions in an appendix to help better understand the model's behavior?
3. Why do you think positions can be predicted in a NAR manner? Isn't it just shifting the burdens to the position predictor? (Since in transformers if it's able to learn positions then it should be trivial to reorder based on those positions)

Minor details:
Some typos need to be fixed.

Overall, it is an interesting idea to predict the positions and word identities separately, and with Gumbel-Softmax and VAE we might be able to optimize the true marginals instead of relying on heuristic search. However, empirically there's been better performance achieved by flow-based models, so I am inclined to reject this work.


--------updates after reading the authors' rebuttal-----
I really appreciate the substantial experiments the authors conducted post-review. However, I still have some concerns regarding the baseline flowseq. Flowseq-large gets better results but is not reported in the main paper. Besides, it is unclear how is the speedup computed since Ma et al 2019 reported that the speedup depends on batch size, and at batch size 120 the speedup is ~X6 for flowseq-base.

Minor issue: in table 1 which dataset are the speedup numbers based on?

**Experience Assessment:**

I have read many papers in this area.

**Review Assessment: Checking Correctness Of Derivations And Theory:**

I carefully checked the derivations and theory.

**Review Assessment: Checking Correctness Of Experiments:**

I carefully checked the experiments.

**Review Assessment: Thoroughness In Paper Reading:**

I read the paper thoroughly.

---

> ### Author Response · Authors · 2019-11-15
> **Response to Reviewer #2 (part #1)**
>
> Thank you for your valuable reviews. We revised accordingly, please check the common response. We have updated our paper and try to address the issues you mentioned in the comment.
>
> Thank you for your valuable reviews. Our paper is heavily updated, please check the common response. We have updated our paper and try to address the issues you mentioned in the comment. We briefly conclude them as below:
>
> Q1: "This work should compare to later baselines such as FlowSeq which gets better performance with flow."
>
> Response:
> - Thanks for your kind reminder.
> - In the latest revision, we have updated the PNAT's result and compare it with the Flowseq[1]. Flowseq can be parallel work released in arxiv on Sept. 5th and published in EMNLP 2019 on Nov. 3rd. ICLR submission was on Sept. 25th. But we're glad to add comparisons. We list the result as follows:
>
>                            Model          |          WMT 14         |   Speedup
>                                                 |    EN-DE      DE-EN |
>   -----------------------------------------------------------------------------------
>                      Transformer     |     27.40       31.33   |   1.0 x
>   -----------------------------------------------------------------------------------
>                                                              w/ KD
>                       Flowseq-base |      21.45       26.16  |    1.1 x
>                      Flowseq-large |      23.72       28.39  |    0.9 x
>                        PNAT               |      23.05       27.18  |    7.3 x
>   -----------------------------------------------------------------------------------
>                                                        w/ Reranking
>    Flowseq-base(LPD n=30) |     23.48        28.4    |    /
>   Flowseq-large(LPD n=30) |     25.31        30.68  |    /
>     PNAT(LPD n=9)                 |     24.46        29.16  |   3.7 x
>
> - As shown in the above, the PNAT training with a strong autoregressive teacher(31.33 BLEU instead of previous 28.47 BLEU in WMT14 DE-EN) achieves a more strong performance with 27.18 BLEU and 7.3 speedups compared to the Transformer.
> - Whether using knowledge distillation or reranking, our PNAT outperforms Flowseq-base. Additionally, the PNAT is faster than Flowseq. As shown in the result, the speedup of PNAT is 7.3 x without reranking, while around the 1.0x from the Flowseq.
>
> Although PNAT does not outperform Flowseq-large, there are some issues should be noted:
> - Both the PNAT and Flowseq-base keep the same setting with previous NAT models and have the same hidden size as 512, but Flowseq-large has a much larger hidden size(2014). The 30.86 BLEU comes from the Flowseq-large which has the larger model setting and select by rescoring from 30 candidates, the result from PNAT still was the base model setting and selected by rescoring from with just 9 candidates the same as previous work[1][2]. With the same model settings of the PNAT and the Flowseq-base, the PNAT achieves a more competitive result than Flowseq (27.18 v.s 26.16 and 29.18 v.s 28.40).
> - Besides, it is also not a completely fair comparison because the Flowseq benefits from the model average strategy which is a kind of ensemble learning while our PNAT is from a single model.
> - Experiments that PNAT with the larger setting is still running. We will update the result once the experiments finished.
>
> Q2: "In table 1, although the proposed approach outperforms imitate-NAT, the speedup is lower, making it hard to judge which is actually better."
>
> Response:
> - There is a trade-off between effectiveness and efficiency. As shown in Table 1,  the effectiveness of the PNAT outperforms the imitate-NAT (27.18 v.s 25.67 and 29.16 v.s 27.28), which indicates that the position is essential for the non-autoregressive generation and consistent with our intuition.
> - The goal of this paper is to propose explicit modeling of positions. It may also be modeled together with imitate-NAT, which we will leave as the future work.

---

> > ### Author Response · Authors · 2019-11-15
> > **Response to Reviewer #2 (part #2)**
> >
> > Q3: "In table 2, why is AR predictor used? What's the performance of NAR predictor? (in general why consider NAR position predictor at all?)"
> >
> > Response:
> > - The PNAT with the autoregressive position predictor is the default setting of our model. We chose it due to it has better performance.
> > - We left the performance and discussion of the NAR position predictor in Section 4.4.
> >
> >          Model                              |               Position Accuracy(%)                 |      WMT14 DE-EN | Speed Up
> >                                                   |   permutation-acc       relative-acc(r=4) |              BLEU        |
> >     --------------------------------------------------------------------------------------------------------------------------------------
> >  Transformer(beam=4)          |                  /                            /                     |             30.68         |     1.0×
> >         NAT-base                         |                 /                             /                     |             16.71         |   13.5×
> >         PNAT w/ HSP                   |             100.00                   100.00               |             46.03         |   12.5×
> >     --------------------------------------------------------------------------------------------------------------------------------------
> >        PNAT w/ AR-Predictor    |             25.30                      59.27                 |              27.11        |     7.3×
> >       PNAT w/ NAR-Predictor  |              23.11                     55.57                 |              20.81         |   11.7×
> >
> > - As we list in the above, PNAT w/ NAR-Predictor still performs better than NAT-base (20.81 v.s 16.71 ) and keep the faster speed (Speedup is 11.7 x).
> >
> > Q4: "It is not clear why heuristic search would work here. Is any pretraining required? Otherwise, since there's no gradient signal for the positions, I'm not sure how the model figures it out."
> >
> > Response:
> > - Heuristic searched position actually provides an association between the source representation and the target word.  PNAT learning to reinforce this word-to-word translation to satisfy this connection.
> > - We do not use any pretraining process here. The parameters of heuristic searching are the encoder and the final word predictor.  The encoder needs to reasonably represent the source input, while the word predictor needs to predict the words from the representation.
> > - The experiment in Section 4.4 has verified this assumption. As shown in Table 3, PNAT w/ HSP completes word-to-word generation well, achieved a very high BLEU score (more than 15.0 BLEU compare to Transformer) which verified that word-to-word connections established through HSP are acceptable.
> >
> > Q5: "How many samples are used in table 1 LPD? Or is it argmax decoding for each length?"
> >
> > Response: We have updated more details in the latest revision. The brief answer is:
> > - In the latest version, we set the \Delta M=4, which means that there are 9 candidates for length parallel decoding (LPD).
> > - Yes. We follow the previous practice[1] and perform the argmax decoding both the position predictor and the decoder for each length.
> >
> > Q6: "Is it possible to include a few examples showing predicted positions in an appendix to help better understand the model's behavior? "
> >
> > Response: Thanks for your suggestion. We have updated the paper and include a few case studies in appendix B.
> >
> > Q7: "Why do you think positions can be predicted in a NAR manner? Isn't it just shifting the burdens to the position predictor? (Since in transformers if it's able to learn positions then it should be trivial to reorder based on those positions)"
> >
> > Response:
> > - It is motivated by the practice of learning syntax position (also called syntax distance). Shen et al[2] have shown that syntax position of words in a sentence could be predicted by neural networks in a non-autoregressive fashion, which even obtained top parsing accuracy among strong parser baselines. We transfer this scenario to the position predictions in NATs.
> > - As shown in Table 3, experiments have shown that the results predicted using the NAR manner are not good enough (20.81 BLEU), which may validate your point of view.
> > - Despite this, it also achieved performance beyond the baseline (16.71 BLEU), so we believe that it is still valuable for exploration.
> >
> > [1] Ma X, Zhou C, Li X, et al. FlowSeq: Non-Autoregressive Conditional Sequence Generation with Generative Flow. EMNLP 2019.
> > [2] Shen Y, Lin Z, Jacob A P, et al. Straight to the tree: Constituency parsing with neural syntactic distance. ACL 2018.

---

### Official Review · AnonReviewer4 · 2019-11-12
**Official Blind Review #4**

**Rating:** 3

**Review:**

This work builds on the non-autoregressive translation (NAT) by using position as a latent variable. Unlike the work by Gu et. al. 2018, where they assume the output word order to follow the word order of the input sentence, this work explores predicting word order supervision as an additional train signal. It shows that predicting the position of the words improves the performance of the translation and paraphrase task. This paper uses a heuristic that the inputs positions and output positions of the decoder with close by embeddings are more likely to represent the position mapping.

Since word order might change across languages the idea of using position based supervision seems promising. The results from the experiments on translation and paraphrasing tasks seem promising as they beat previously established baselines. With techniques like length parallel decoding from Gu et. al. 2018, PNAT performs much better than the baselines. For the paraphrasing task, it is interesting to observe that PNAT beats the autoregressive transformer based model.

Pros:
- This work gives a convincing argument to model word position prediction as a latent variable.
- Experiments show PNAT beats baseline models for Translation and Paraphrasing task.
- It also shows that using position supervision increases the convergence speed of the model.

Questions:
- While position prediction seems like a good idea, I am not fully convinced of the heuristic used -  similarity between input (d_i) and output (y_j) is used to determine the position supervision. Since (d_i)s undergo many transformations to produce (y_j)s, the embedding vectors don't necessarily have to be similar for, some d_i to have greater influence on a particular y_j. It would be nice to verify this assumption using some gold data or some manual checks.

- In addition to the previous point, is the model pertained before this heuristic is used? Since, starting with random initialization might just reinforce random position mappings based on initial conditions.

-  In describing the HSP, could you please make it more clear how the z_i are decided? Is it that the iteratively best (d_i, y_j) is selected as the z_i and then d_i & y_j are removed from the corresponding sides?

- The tables assume that the reader knows about the abbreviations. Could you please add what NPD, WT, etc. mean?

- It would be nice to see the respective results with NAR position predictor since the discussion is about building a Non Autoregressive model.

- Are the gains from NAT lost by using AR position predictor since autoregressive prediction is added indirectly to the whole model?

- In Table 2 PNAT w/HSP seems to have amazing performance compared to other models. Could the authors shed some light on why this cannot be used directly? Is it because of delays due to the iterative process in extracting z_i?



Minor:

- There are a couple of typos.



**Experience Assessment:**

I have read many papers in this area.

**Review Assessment: Checking Correctness Of Derivations And Theory:**

I assessed the sensibility of the derivations and theory.

**Review Assessment: Checking Correctness Of Experiments:**

I assessed the sensibility of the experiments.

**Review Assessment: Thoroughness In Paper Reading:**

I read the paper at least twice and used my best judgement in assessing the paper.

---

> ### Author Response · Authors · 2019-11-15
> **Response to Reviewer #4 (Part #1 )**
>
> Thank you for your valuable reviews. We revised accordingly, please check the common response. We have updated our paper and try to address the issues you mentioned in the comment.
>
> Q1: "While position prediction seems like a good idea, I am not fully convinced of the heuristic used -  similarity between input (d_i) and output (y_j) is used to determine the position supervision. Since (d_i)s undergo many transformations to produce (y_j)s, the embedding vectors don't necessarily have to be similar for, some d_i to have greater influence on a particular y_j. It would be nice to verify this assumption using some gold data or some manual checks."
>
> Response:
> - Thanks for your question.
> - Actually, we have verified this assumption in Section 4.4. As shown in Table 3 we list as below, PNAT w/ HSP has achieved a promising result with a large margin than Transformer (46.03 v.s 30.68, more than 15 BLEU), which shows the HSP is very effective.
>
>
>          Model                              |               Position Accuracy(%)                 |      WMT14 DE-EN | Speed Up
>                                                   |   permutation-acc       relative-acc(r=4) |              BLEU        |
>    --------------------------------------------------------------------------------------------------------------------------------------
> Transformer(beam=4)          |                  /                            /                     |             30.68         |     1.0×
>         NAT-base                         |                 /                             /                     |             16.71         |   13.5×
>         PNAT w/ HSP                   |             100.00                   100.00               |             46.03         |   12.5×
>    --------------------------------------------------------------------------------------------------------------------------------------
>        PNAT w/ AR-Predictor    |             25.30                      59.27                 |              27.11        |     7.3×
>       PNAT w/ NAR-Predictor  |              23.11                     55.57                 |              20.81         |   11.7×
>
> Table 3: Results on the validation set of WMT14 DE-EN with different position strategy. “HSP” means the reference position sequence derived from the heuristic position searching.
>
> Q2: "In addition to the previous point, is the model pertained before this heuristic is used? Since, starting with random initialization might just reinforce random position mappings based on initial conditions."
>
> Response:
> - We do not use any pretraining process here.
> - In fact, the parameters used in heuristic searching are the encoder and the word predictor, which is also part of the NAT models. The encoder needs to reasonably represent the source input, while the word predictor needs to predict the words from the representation.
> - With the training of the model, the encoder and word predictor is updated for its goals, which leads the PNAT can not trivially reinforce random position mappings just based on random initial conditions.
>
> Q3: "In describing the HSP, could you please make it more clear how the z_i are decided? Is it that the iteratively best ( d_i, y_j ) is selected as the z_i and then d_i & y_j are removed from the corresponding sides?"
>
> Response:
> - Yes. For HSP, we chose an approximation algorithm that iteratively determines the position pair. At each step, the best matching pair of d_i and y_j are selected and then removed d_i & y_j from the set of candidate matches.
> - We chose this approach because it is easier to perform batch computations for training than the Hungarian algorithm.
>
> Q4: "The tables assume that the reader knows about the abbreviations. Could you please add what NPD, WT, etc. mean?"
>
> Response: Noted for thanks. We addressed this in the latest revision.
> - NPD: noisy parallel decoding used for beam searching in previous NATs work, such as Gu et al.[1]
> - WT: weak teacher version of Transformer which has a similar performance to the Transformer used in Gu et al.
> - It is used for generating the distillation data or rescoring the output sequences.
>
> Q5: "It would be nice to see the respective results with NAR position predictor since the discussion is about building a Non Autoregressive model."
>
> Response:
> - Thanks for your suggestion. We initially try to use a NAR position predictor but do not get enough position accuracies.
> - The performance with NAR-Predictor is not good enough, we thus chose to use AR-Predictor, which didn't drop much in speed (11.7x to 7.3 x) and had significant performance benefits(20.81 BLEU to 27.11 BLEU).
> - The current PNAT model predicts the word non-autoregressively while based on the autoregressive position predictor has achieved a strong performance(29.16 BLEU and 3.7 Speedup).
> - We left the exploration of NAR-Predictor in the future work which is still promising.

---

> > ### Author Response · Authors · 2019-11-15
> > **Response to Reviewer #4 (Part #2)**
> >
> > Q6: "Are the gains from NAT lost by using AR position predictor since autoregressive prediction is added indirectly to the whole model?"
> >
> > Response:
> > - Gains from NAT are not lost by using the AR position predictor. The word prediction of the PNAT is still with non-autoregressively fashion. The PNAT with AR-predictor still has beneficial in efficiency compared to the Transformer(The speedup compared with AT model is 7.3 x ) due it has a smaller search space.( O(Sentence_Length) v.s O(Vocab_size) ).
> > - We also conduct the experiment in Section 4.4 to evaluate the PNAT using the NAR position predictor. PNAT with NAR-Predictor achieves a faster speedup than PNAT w/ AR-Predictor (11.7x v.s 7.3x),  however, it comes at the performance degradation ( 20.81 BLEU v.s 27.11 BLEU ).
> > - Beside, PNAT w/ HSP has achieved a very promising result, which means that if the performance of position prediction can be higher, the quality of its generation will be better. This is still very impressive and attractive.
> >
> > Q7: "In Table 2 PNAT w/HSP seems to have amazing performance compared to other models. Could the authors shed some light on why this cannot be used directly? Is it because of delays due to the iterative process in extracting z_i?"
> >
> > Response:
> > - HSP stands for the reference position sequence we used for training, it needs to take the decoder inputs and the reference as the input. We conduct this experiment to verify the effectiveness of the heuristics searched position.
> > - PNAT w/HSP indicates the oracle performance in the current setting. It is naturally cannot be used at the inference stage because it takes the reference as the input. Not due to the iterative process in extracting z_i.
> >
> > [1] Gu J, Bradbury J, Xiong C, et al. Non-autoregressive neural machine translation. ICLR 2018.

---

### Public Comment · ~Kaaliya_Budhil2 · 2019-10-13
**Missing baseline results**

NAT-Reg gets a BLEU score of 28.90 on WMT14 De-En with an AT teacher in similar performance as you used (31.29 vs 31.25), which is omitted in the lower half in Table 1. Instead, only the weak teacher version is reported, which utilized an AT teacher with only 28.76 BLEU score. I believe it is necessary to keep the performance of the AT teacher consistent to construct a fair comparison.

---

> ### Author Response · Authors · 2019-10-21
> **Response to "Missing baseline results"**
>
> Thank you very much for your comments!  NAT-REG is a very strong work in non-autoregressive Transformer (NAT), but we fairly compare PNAT with NAT-REG. We will clarify that to avoid misunderstanding.
>        1. The performance of PNAT's AT teacher is 28.47 BLEU, instead of 31.25 BLEU.
>        As we mentioned in the paragraph "Knowledge Distillation" of Section 4.2, the distillation data we used is from https://github.com/nyu-dl/dl4mt-nonauto released by Lee et al. 2018 [1] (AT teacher with 28.47 BLEU), which has the similar KD performance to most of  the previous NAT work, such as NAT-FT, IR-NAT, ENAT, and NAT-REG (WT). The Transformer model (31.25 BLEU) we listed in Table 1 is just for reranking the candidate instead of as an AT teacher. Thus we do not unfairly compare with NAT-REG(WT) by using a better AT teacher.
>        2.  PNAT vs. NAT-REG without reranking.
>        Our experimental setting is closer to NAT-REG-WT (AT teacher with 28.76) instead of NAT-REG (AT teacher with 31.29). Under this setting, our proposed PNAT without reranking achieves the performance of 26.65 BLEU, and the NAT-REG (WT) is 23.20. Even with a strong teacher model (31.29), the NAT-REG achieved only 24.77 (improved by 1.57 with NAT-REG with a weaker teacher), which is still lower than PNAT (26.65) with a weaker teacher (28.47).
>        3. PNAT v.s NAT-REG with a strong AT teacher (31.29).
>        Thanks for your kind notes. We will add NAT-REG in reranking results. Benefit from reranking, NAT-REG achieves the performance of 28.90 BLEU with the setting of (31.29 AT teacher, 31.29 AT reranker, candidate=9). It should be noted that a direct comparison between NAT-REG and PNAT is unfair because PNAT's AT teacher gives much lower BLEU than NAT-REG's, which is the upper bound of the final performance. Thanks for your suggestion, we will also add the result of PNAT with a stronger setting (31.25 AT teacher, 31.25 AT reranker, candidate=9) in the next revision for a thorough comparison.
>
> [1] Deterministic Non-Autoregressive Neural Sequence Modeling by Iterative Refinement, ACL 2018.

---

### Public Comment · ~Aurko_Roy1 · 2019-10-14
**Missing references**

This work is missing the following references on non-autoregressive neural machine translation using latent variables:

https://arxiv.org/abs/1803.03382
https://arxiv.org/abs/1805.11063

---

> ### Author Response · Authors · 2019-10-20
> **Response to "Missing reference"**
>
> Thanks for your kind reminder!  We will cite them and give some comparisons.
> The mentioned two papers are related to our submission, which also uses the latent variable for non-autoregressive NMT. The two papers utilize discrete latent variables for making decoding more parallelizable, in which the latent variable is the continuous semantic vector with no explicit meaning. But our proposed PNAT utilizes the latent variable to model the inherent order of the decoding outputs for non-autoregressive decoding, in which the latent variable is the position of words in the output sentence.
>
> [1] Fast Decoding in Sequence Models Using Discrete Latent Variables, ICML 2018.
> [2] Theory and Experiments on Vector Quantized Autoencoders, https://arxiv.org/abs/1805.11063

---

### Public Comment · ~Jiawei_Zhou1 · 2019-10-24
**Experiments on More Data**

To be consistent with the literature and for comparison reasons, are there any experimental results for WMT14 En-De direction, as well as on WMT16 Ro-En and En-Ro translations? Currently they seem to be missing from the paper.

---

> ### Author Response · Authors · 2019-11-02
> **Response to "Experiments on More Data"**
>
> Thank you very much for your comments! We will add experiments on more data for comparison.

---

### Public Comment · ~Xuezhe_Ma1 · 2019-11-05
**missing references**

This work is missing the following reference on non-autoregressive neural machine translation:
https://arxiv.org/abs/1909.02480

---

> ### Author Response · Authors · 2019-11-15
> **Response to "Missing Reference"**
>
> Thank you very much for your comments! We have updated the paper and added it.

---

### Public Comment · ~Chenze_Shao2 · 2019-11-07
**missing references**

This work is missing the following references on non-autoregressive neural machine translation:
https://arxiv.org/abs/1906.09444

---

> ### Author Response · Authors · 2019-11-15
> **Response to "Missing Reference"**
>
> Thank you very much for your comments! We have updated the paper and added it.

---

### Author Response · Authors · 2019-11-15
**Common Response**

We thank all the reviewers for their insightful comments. We have revised the paper and add more experiments on WMT EN->DE direction.

We also change the KD data from higher performance Transformer and keep the same with recent NAT papers, including imitate-NAT[1], NAT-REG[2] and ENAT[3].  We achieved a significantly better result on WMT14 DE-EN from 27.9 to 29.18.

[1] Wei B, Wang M, Zhou H, et al. Imitation Learning for Non-Autoregressive Neural Machine Translation. ACL 2019.
[2] Wang Y, Tian F, He D, et al. Non-autoregressive machine translation with auxiliary regularization. AAAI 2019.
[3] Guo J, Tan X, He D, et al. Non-autoregressive neural machine translation with enhanced decoder input. AAAI 2019.

---

### Decision · Program_Chairs · 2019-12-19

**Decision:**

Reject

**Comment:**

This paper presents a non-autoregressive NMT model which predicts the positions of the words to be produced as a latent variable in addition to predicting the words. This is a novel idea in the field of several other papers which are trying to do similar things, and obtains good results on benchmark tasks. The major concerns are systematic comparisons with the FlowSeq paper which seems to have been published before the ICLR submission deadline. The reviewers are still not convinced by the empirical performance comparison as well as speed comparisons. With some more work this could be a good contribution. As of now, I am recommending a Rejection.